# Effects of Protein-Chelated Zinc Combined with Mannan-Rich Fraction to Replace High-Dose Zinc Oxide on Growth Performance, Nutrient Digestibility, and Intestinal Health in Weaned Piglets

**DOI:** 10.3390/ani12233407

**Published:** 2022-12-02

**Authors:** Gang Zhang, Jinbiao Zhao, Gang Lin, Yuhan Guo, Defa Li, Yi Wu

**Affiliations:** 1State Key Laboratory of Animal Nutrition, College of Animal Science and Technology, China Agricultural University, Beijing 100193, China; 2Nutrition Laboratory of Wellhope Foods Co., Ltd., Shengyang 110164, China; 3Key Laboratory of Agrifood Safety and Quality, Institute of Quality Standards and Testing Technology for Agricultural Products, Chinese Academy of Agricultural Sciences, Ministry of Agriculture and Rural Affairs, Beijing 100081, China

**Keywords:** protein-chelated zinc, mannan-rich fraction, zinc oxide, weaned piglets, growth performance, gut microbiota

## Abstract

**Simple Summary:**

The early weaning of piglets is a key strategy for the modern pig industry. However, the digestive system of early-weaned piglets is not well developed, and weaned piglets suffering from psychological, nutritional and environmental stresses, cause significant losses to the pig industry. In order to mitigate the challenges of weaning, adding zinc oxide to the diet of weaned piglets has become standard in the industry. However, the excessive output of zinc from pigs’ feces to the water and soil will produce toxicity to crops and threaten animal and human health. Mannan-rich fraction, derived from *Saccharomyces cerevisiae*, can improve gut health and the immune system. In addition, protein-chelated zinc has stable chemical properties, a higher absorption rate and biological potency. Therefore, we speculate that the combination of protein-chelated zinc and mannan-rich fraction will be an alternative to high-dose zinc oxide to relieve weaning stress. Our study showed that the combined addition of protein-chelated zinc and mannan-rich fraction is an alternative for zinc oxide in beneficially supporting the growth performance and intestinal health of weaned piglets, as well as contributing to a lower diarrhea rate and environmental pollution from fecal zinc excretion.

**Abstract:**

A total of 168 weaned piglets (average initial body weight of 7.70 ± 0.75 kg) were used in a 4-week feeding trial to investigate the effects of dietary supplementation with protein-chelated zinc (Zn-Pro) alone or combined with a mannan-rich fraction (MRF) to replace high-dose zinc oxide (ZnO) for weaned piglets. The dietary treatments included a basal diet as control (CON), a ZnO diet (basal diet + 1600 mg Zn/kg from ZnO), a Zn-Pro diet (basal diet + 60 mg Zn/kg from Zn-Pro), and a MRF plus Zn-Pro diet (MRP, basal diet + 800 mg/kg MRF + 60 mg Zn/kg from Zn-Pro). The average daily gain of piglets in the MRP group was higher (*p* ≤ 0.05) than that in CON and Zn-Pro groups during d 15–28 and d 1–28 of experiment. The apparent total tract digestibility of dry matter, organic matter, and crude protein in the MRP group was higher (*p* ≤ 0.05) than that in the CON group. The serum insulin-like growth factor-1 level in the MRP group was markedly higher (*p* ≤ 0.05) than that of piglets in the other three treatment groups. Piglets fed the Zn-Pro and ZnO diets had greater (*p* ≤ 0.05) acetic acid in cecal digesta than those fed the CON diet, while the MRP diet had higher (*p* ≤ 0.05) cecal propionate concentration than those that were fed the CON diet on d 28 of experiment. Moreover, the villus height of ileum in the MRP group tended to be greater than the CON group (*p* = 0.09). Compared with the CON and MRP groups, the relative abundance of Lactobacillaceae (*p* = 0.08) and Lachnospiraceae (*p* = 0.09) in the Zn-Pro group showed an increasing trend. The relative abundance of Prevotellaceae in the Zn-Pro group was significantly lower (*p* ≤ 0.05) than that in the MRP group. In conclusion, the combined addition of MRF and Zn-Pro acted as a suitable alternative to ZnO to beneficially support the growth performance and intestinal health of weaned piglets, as well as contribute to a lower diarrhea rate and environmental pollution from fecal zinc excretion.

## 1. Introduction

The weaning of piglets in the early period is important in pig production. It can not only improve the production performance of piglets and the utilization rate of pens, but also reduce the probability of sow-to-pig disease transmission [1]. However, the intestinal system of early-weaned piglets is not well developed, and the secretions of endogenous enzymes and gastric acid are insufficient, so the nutrients in the feed cannot be effectively digested and utilized [2]. In addition, suffering from environmental, nutritional and psychological stresses after weaning, piglets often show low feed utilization, diarrhea, and stunted growth, which cause significant losses to the pig industry [2]. In recent years, in order to mitigate the challenges of weaning as well as improve the growth performance and gut health of piglets, adding zinc oxide (ZnO) (Zn ≥ 2000 mg/kg) with pharmacological doses to the diet of weaned piglets has become standard in the industry [3,4]. However, long-term high-dose Zn from ZnO will eventually accumulate in soil, causing heavy metal contamination and increasing microbial resistance, which will potentially harm the health of animals and humans [4,5,6]. Excessive zinc (Zn) content (2000–3000 mg/kg) in the diet also reduces the bioavailability of other mineral elements and increases feed costs [7]. Due to environmental considerations, China currently allows 1600 mg/kg of additional Zn in weaned piglet diet in only the first two weeks, and the European Union will ban the use of pharmacological doses of Zn that are greater than 150 mg/kg zinc in piglet feed starting in June 2022 [8,9]. Therefore, it is important to develop ZnO replacement products to aid the weaning transition to help with enhanced growth in weaned piglets under low zinc conditions, and to ensure the sustainable and healthy development of the pig industry [10].

With reduced zinc excretion, the improvement of growth performance and the reduction of diarrhea rate are the main indicators for evaluating alternatives to high-dose zinc oxide. Mannan oligosaccharide (MOS) products from *Saccharomyces cerevisiae* cell wall, have attracted increasing attention in animal husbandry because it can improve gut health and the immune system, thereby boosting animal growth performance [11]. Similar to the effect of ZnO in high dose, MOS inhibits the proliferation of harmful gut bacteria (such as *Escherichia coli* and *Salmonella* spp.), regulates the composition of gut flora, and then reduces the diarrhea rate of pigs [12,13]. Research showed that MOS and ZnO in high dose had synergistic effects in promoting the growth performance of pigs, and MOS can also reduce environmental pollution caused by poor retention rates [12,14,15]. A mannan-rich fraction (MRF) can improve gut health and the immune system and has been launched as the second generation of these MOS products [16]. Compared with ZnO, protein-chelated zinc (Zn-Pro) has stable chemical properties, a higher absorption rate, and biological potency, so it has broad application [17].

Therefore, we conducted this study to evaluate the effects of the combination of Zn-Pro and MRF, which may serve as alternatives to ZnO, on the growth performance, diarrhea score, nutrient digestibility, zinc excretion, serum hormone levels, gut morphology, gut microbiota, and volatile fatty acids (VFAs) of weaned piglets.

## 2. Materials and Methods

The protein-chelated zinc (Bioplex^®^ Zinc, Zinc ≥ 15%) and MRF (composed from *Saccharomyces cerevisiae* cell wall and brewer’s dry yeast, MRF ≥ 108 g/kg) used in the test were provided by Beijing Alltech Biological Products Co., Ltd. (Beijing, China) and ZnO was commonly used feed grade Zn.

### 2.1. Animals and Experimental Design

One hundred and sixty-eight (half male and half female) healthy pigs [Duroc × (Landrace × Yorkshire)] were weaned at d 21 of age, then fed a commercial creep feed for 7 days and transferred to a nursery facility on d 28 of age to conduct the 28-day experiment. All pigs were routinely immunized according to the farm immunization program. With average initial body weight of 7.70 ± 0.75 kg, pigs were randomly allocated into four groups with seven pens and six piglets per pen. The dietary treatments included a basal diet as control (CON), a ZnO diet (basal diet + 1600 mg Zn/kg from ZnO), a Zn-Pro diet (basal diet + 60 mg Zn/kg from Zn-Pro), and a MRF plus Zn-Pro diet (MRP, basal diet + 800 mg/kg MRF + 60 mg Zn/kg from Zn-Pro). The experiment used a low-protein corn-soybean meal basal diet with synthetic amino acids. The basal diet except zinc was formulated according to the nutritional requirements recommended by NRC (2012). The nutritional level and composition of the basal diet are shown in Table 1. The feeding experiment lasted for 28 days. All diets were supplemented with 0.3% chromium trioxide (Cr_2_O_3_) as an exogenous indicator for determining nutrient digestibility. All piglets were fed in 1.2 × 2.1 m^2^ pens with plastic slatted floors in a room. Pigs ad libitum accessed feed (in a mash form) and water. The relative humidity in the nursery room was kept at 65–75%, and the room temperature was adjusted at 27 °C during the first week and gradually decreased by 1 °C per week.

### 2.2. Sample Collection

The feed consumption of each pen and individual piglet weights were recorded on d 1, 14, and 28 of experiment to measure average daily gain (ADG), average daily feed intake (ADFI) and feed-to-gain ratio (F:G). Diarrhea and fecal consistency were recorded every morning and afternoon. A grading system was used to determine the diarrhea severity and presence: 1 = hard solid feces; 2 = slightly soft feces; 3 = soft, partially formed feces; 4 = loose, semiliquid feces (mild diarrhea); and 5 = watery, mucous-like feces (severe diarrhea). The diarrhea incidence was defined in feces that were semiliquid, watery, and mucous-like (4 and 5 score), and the diarrhea incidence was measured: Diarrhea incidence (%) = the total number of diarrhea piglets/(total number of piglets × total observational days) × 100(1)

Feed samples (500 g) in each group were collected before the start of the experiment. On d 25~27 of experiment, fresh fecal grab samples (300 g) from all 6 piglets in each pen were collected, and then pooled and mixed. The fecal samples were oven-dried at 65 °C for 72 h. The fecal samples and feed were ground by a mini-type hammer no-sieve disintegrator, and then passed through a 1 mm sieve.

On d 14 and d 28 of experiment, one barrow (with average body weight) from each pen was selected to collect blood. The blood sample was rested for 45 min at room temperature prior to serum separation. Serum was collected after centrifugation at 3000× *g* for 10 min at 4 °C and stored at −20 °C.

At the end, pigs supplying blood samples were euthanized and necropsied, and then the intestine was removed for sample collection. The middle parts of the ileum and jejunum were cleaned and cut down the 2 cm gut segment, and then fixed in 10% formalin. In addition, colon and cecal contents from each pig were aseptically collected and immediately stored at −80 °C for detection of microbial composition and VFA levels. 

### 2.3. Analytical Methods

Feces from d 28 of experiment were thawed naturally, and 0.5 g was weighed. DNA was extracted using DNA Extraction Kit (Omega, USA). 16S rRNA gene in V3–V4 region was amplified using the specific primers 338F (5′-ACTCCTACGGGAGGCAGCAG-3′) and 806R (5′-GGACTACHVGGGTWTCTAAT-3′). Amplicon libraries were separated by agarose gel electrophoresis and purified. Purified amplicons were sequenced using Illumina MiSeq platform (San Diego, CA, USA). Microbial sequencing data were subjected to bioinformatics analysis. QIIME (version 2.0: https://qiime2.org, accessed on 10 December 2019) was used to demultiplex and quality-filter raw FASTQ files format. The sequences were clustered into operational taxonomic units (OTUs) with UPARSE (version 7.0.1001: http://drive5.com/uparse/, accessed on 13 December 2019) with a novel ”greedy” algorithm that performs chimera filtering and OTU clustering simultaneously and the identity threshold was set at 97%. 

The VFA concentrations in chyme were determined with reference to the improved method [18]. Briefly, frozen samples were thawed at room temperature, about 0.5 g of fecal sample was weighed and diluted with 8 mL ultrapure water. After 30 min of sonication, the sample solution was centrifuged at 4000× *g* for 10 min. The supernatant was diluted 50-fold in ultrapure water and then filtered through a 0.22 mm membrane filter. The VFA concentrations were determined using HP 6890 ion gas chromatography (Agilent, Santa Clara, CA, USA).

The intestinal tissue samples were fixed in formalin for 24 h, embedded in paraffin, and sectioned. Finally, the tissue was stained using hematoxylin-eosin (HE). Twelve well-oriented intact villi and their crypts per section were measured using image analyzer (Leica Application Suite; Leica, Wetzlar, Germany). 

After the samples were wet-digested with a nitric acid-perchloric acid (3:1) mixture and diluted with deionized distilled water, the concentration of chromium and zinc was determined by flame atomic absorption spectrometry (AAnalyst 400, Perkins Elmer, Waltham, MA, USA). Fecal samples and feed were analyzed for dry matter (DM; method GB/T 6435-2014), crude protein (CP; method GB/T 6432-1994), and ash (method GB/T 6438-2007). Gross energy (GE) was measured by an automatic adiabatic oxygen bomb calorimeter (Parr 6400, Calorimeter, Moline, IL, USA). Organic matter (OM) was calculated as the difference between ash and DM. Feed samples were analyzed for acid detergent fiber (ADF) and neutral detergent fiber (NDF) according to van Soest et al. [19]. The apparent total tract digestibility (ATTD) was calculated by the following equation:ATTD (%) = 100 − (Cr_diet_ × N_feces_)/(Cr_feces_ × N_diet_) × 100
where Cr_diet_ represents the chromium concentration in the diet (g/kg), N_feces_ is the nutrient concentration in feces (g/kg), Cr_feces_ is the chromium concentration in feces (g/kg), and N_diet_ is the nutrient concentration in the diet (g/kg).

Serum growth hormone (GH), insulin-like growth factor-I (IGF-I) and ghrelin were measured using ELISA kits (Beijing Sino-UK Institute of Biological Technology, Beijing, China) according to the manufacturer’s instruction.

### 2.4. Statistical Analysis

Data were checked for normality and outliers using the UNIVARIATE procedure of SAS 9.4 (SAS Institute Inc., Cary, NC, USA). Pen was used as an experimental unit for analyzing the nutrient digestibility data, incidence of diarrhea, and growth performance, whereas each individual pig was used as the experimental unit for the other parameters. Analysis of variance was performed with the GLM model. Mean values of treatment groups were calculated with LSMEANS method multiple comparison, and multiple comparisons were performed with the Student-Newman-Keuls procedure. For microbial sequencing data, R tools were used to generate community figures with the data from document “tax.phylum.xls, tax.family.xls, and tax. genus.xls”. The bar figures of bacterial community were conducted with R ggplot package and heatmaps were conducted with R vegan package. The abundance of microbiota was compared with Kruskal-Wallis test. Statistical significance was considered with *p* ≤ 0.05, and *p* values between 0.05 and 0.10 were considered as a tendency.

## 3. Results

### 3.1. Performance and Diarrhea

The diarrhea rate and growth performance of weaned piglets were shown in Table 2. The F:G was not different (*p* > 0.05) among all groups. Diarrhea incidence in the CON treatment was higher (*p* ≤ 0.05) than that in the ZnO, Zn-Pro and MRP treatments throughout the trial. In addition, during d 15 to 28 of experiment and the whole period, the ADG in the MRP group was increased (*p* ≤ 0.05) compared with the CON and Zn-Pro groups. During d 15 to 28 of experiment, the ADFI in the MRP and ZnO groups trended to be higher (*p* = 0.10) than that in the CON and Zn-Pro groups. At the end of the experiment, the body weight in the MRP treatment tended to be higher (*p* = 0.07) than that in the CON treatment. 

### 3.2. Fecal Zinc Concentration and Nutrient Digestibility

The nutrient apparent digestibility and fecal zinc excretion were shown in Table 3. The MRP group had higher (*p* ≤ 0.05) ATTD of CP, DM and OM compared with those in the CON group. Additionally, the MRP group tended to increase (*p* = 0.09) the ATTD of GE compared with that in the CON group. The Zn concentration in the feces was decreased (*p* ≤ 0.05) in the Zn-Pro and MRP groups compared with the ZnO group. However, the Zn concentration in the feces was increased (*p* ≤ 0.05) in the Zn-Pro and MRP groups compared with the CON treatment.

### 3.3. Serum Growth-Related Hormone

The serum growth-related hormone levels of GH, IGF-1 and ghrelin were shown in Table 4. The serum IGF-1 was increased (*p* ≤ 0.05) in the MRP group compared with that in the CON, ZnO, and Zn-Pro groups on d 14. On d 28, piglets fed the ZnO diet exhibited increased (*p* ≤ 0.05) GH level in the serum unlike those in the Zn-Pro treatment.

### 3.4. Intestinal Morphology

In Table 5, indicators of intestinal morphology showed no significant differences (*p* > 0.05) among all groups. The ratio of villus height to crypt depth in the MRP treatment tended to be higher (*p* = 0.09) than that in the CON treatment. 

### 3.5. Intestinal Volatile Fatty Acid Concentration

The VFA concentrations were shown in Table 6. Piglets in the Zn-Pro group had an increased (*p* ≤ 0.05) acetic acid concentration in cecal digesta compared with those in the CON and MRP groups, and had a significantly increased (*p* ≤ 0.05) total VFA concentration compared with those in the CON group. Piglets in the ZnO group had a significantly increased (*p* ≤ 0.05) cecal acetic acid content compared with those in the CON group. The propionic acid concentration in cecal digesta of piglets in the MRP group was significantly higher (*p* ≤ 0.05) than that in the CON and Zn-Pro groups.

### 3.6. Microbiota Community

The average sequence length of the species was 439 bp. A total of 1277 OTUs were identified and classified into 25 phyla, 35 classes, 67 orders, 120 families, 287 genera, and 489 species. The Ace, Chao, and Shannon indicating colonic microbial community diversity and richness had no difference among the four groups (*p* > 0.05, Figure 1). The relative abundance of the colonic microbial communities was shown in Table 7. The dominant taxa in all groups were Ruminococcaceae, Clostridiaceae_1, Prevotellaceae, Lactobacillaceae, Erysipelotrichaceae and Lachnospiraceae, which accounted for about 80% of the microbial sequences. The colonic microbiota of piglets in the ZnO group tended to have lower (*p* = 0.09) Ruminococcaceae abundance than that in the CON group. Compared with the CON and MRP groups, the relative abundance of Lactobacillaceae (*p* = 0.08) and Lachnospiraceae (*p* = 0.09) abundance in the Zn-Pro group tended to increase. In addition, Prevotellaceae in the Zn-Pro treatment group was lower than that in the MRP group (*p* ≤ 0.05). At the genus level, the colonic microbiota of piglets in ZnO and Zn-Pro groups tended to have lower (*p* = 0.06) abundance of *Ruminococcaceae_UCG-005* compared with that in the CON group. The relative abundance of *Prevotellaceae_NK3B31_group* (*p* = 0.08) and *Prevotella_2* (*p* = 0.09) tended to be lower in piglets fed Zn-Pro than those in the MRP group.

## 4. Discussion

After weaning, pigs typically experience a period of growth lag caused by stresses from the changes in nutrition, psychology, and environment [20]. Pharmacological doses of ZnO can increase feed intake by promoting the secretion of ghrelin and pancreatic digestive enzymes [21,22], thereby improving growth performance and alleviating negative effects associated with delayed growth in post-weaned piglets, which includes the thinning of intestinal mucosa, decrease in immunity and reduction in the expression of tight junction proteins [23]. Although the mechanisms are different, both ZnO and MOS have the potential to alleviate weaning stress. Yeast-derived MOS promotes animal growth performance mainly by preventing pathogen adhesion, as well as improving intestinal microbial community structure and immune function [11,24]. Furthermore, our previous study found that MRF could partially replace ZnO by lowering piglet diarrhea rates, as well as improving nutrient digestibility and growth performance [12]. Therefore, this study was conducted to evaluate whether MRF plus zinc proteinate could replace high-dose ZnO on the supporting piglet growth performance and intestinal health, and, moreover, reduce Zn excretion.

All treatments resulted in equal performance to the control during the first two weeks, but ADG in the second period and overall period during post-weaning was greater with the combination use of ZnO and MRF. This result was consistent with the previously reported growth-promoting effect of the combined use of ZnO and MRF [12]. Castillo et al. reported that the combined addition of MOS and organic zinc could improve the feed conversion efficiency of piglets in the first two weeks after weaning [25]. Studies found that adding MOS-rich yeast products (yeast or yeast enzymolysate) and ZnO to the diet improved the growth performance and immune function of weaned piglets [10,26]. Similar to the results of this study, low doses of ZnO (1000–1500 mg Zn/kg) [10,21], and even lower doses of coated ZnO (500 mg Zn/kg) also improved the growth performance of piglets [27]. However, some studies reported that 1600 mg Zn/kg ZnO failed to promote growth performance in piglets [12,28]. These different results in studies might be due to the age and health status of piglets and the different feeding managements. Recent studies have shown that the supplemental dose of ZnO in the diet reaches 1400 mg/kg in the first two weeks after weaning, that is, the daily intake of zinc is 400 mg, could reduce the risk of growth retardation and diarrhea rate in piglets [29]. Therefore, despite Zn-Pro with higher bioavailability, the low supplemental dose (60 mg Zn/kg) may be part of the reason why Zn-Pro failed to improve the growth performance of piglets in this study [16,30,31]. However, at the same dose of Zn as the Zn-Pro group, the combined addition of Zn-Pro and MRF improved growth performance, which may be related to the effect of MRF on intestinal health. Research has shown that MRF improved the diversity and richness of intestinal microbiota in pigs, and maintained the stability of the intestinal ecosystem [12]. In our study, there was no difference in the diversity and richness of intestinal microbiota among different groups, which might be because the effect of MRF on gut microbes was attenuated by Zn-Pro.

After early weaning, pigs are prone to diarrhea under weaning stresses due to poor digestive function [2]. The results of this study showed that ZnO treatment at 1600 mg Zn/kg significantly lowered the incidence of diarrhea in piglets, which was consistent with the previous studies [12,21,28]. In practical production, nutritional strategies (such as the addition of ZnO or prebiotics) promote the gut development and lessen the diarrhea of piglets [32]. Similarly, Castillo et al. [25] reported that dietary supplementation with organic zinc or MOS improved the fecal scores of piglets. Zhang et al. [12] also found that 800 mg/kg MRF had the same anti-diarrheal effect as 1600 mg Zn/kg ZnO. In this study, the Zn-Pro alone or in combination with MRF to the diet resulted in a lower diarrhea rate in piglets at a level comparable to that of the ZnO group. The anti-diarrheal effect of MRF is mainly attributed to its effect on inhibiting intestinal inflammation and pathogen colonization (such as *Escherichia coli* and *Salmonella* spp.), thereby reducing the risk of diarrhea [13,33]. 

The increase of villus height and the decrease of crypt depth can enlarge the intestinal epidermal area, and, thus, promote the absorption efficiency of nutrients in the small intestine [34]. Therefore, villus height and crypt depth are important indicators to evaluate the intestinal morphology. Weaning stress causes damage to gut morphology, including villus atrophy and crypt hyperplasia, and disrupts gut barrier function in pigs [34]. In addition, this study found that the combination of organic zinc with MOS in diet lowered the crypt depth of jejunum and raised the villus height/crypt depth ratio, but the additions of organic zinc or MOS alone did not affect the gut morphology [25]. The beneficial effects of MOS on gut are attributed to the regulation of gut microbiota and the reduction of the pathogenic bacterial load, thus, improving intestinal morphology [35]. Studies have confirmed that MOS and/or ZnO in the diet improved the intestinal morphology of piglets and increased the digestibility of nutrients [12,27,36]. At the same time, organic zinc has been shown to improve intestinal morphology and increase the absorption area of nutrients by promoting intestinal epithelial cell proliferation and protein synthesis [25]. Long et al. [26] reported that the combined use of MOS-rich yeast and ZnO improved the digestibility of GE, OM and CP. In this study, the ileum villus height tended to increase and the ATTD of DM, OM, and CP were significantly increased in the MRP group. Therefore, the combination of Zn-Pro and MRF in place of high levels of ZnO might benefit the intestinal morphology and, thus, improve the nutrient digestibility of pigs.

Ghrelin is a peptide hormone which stimulates GH secretion from the pituitary gland. Ghrelin and GH regulate IGF-1 synthesis, and this GH/IGF-1 axis, which relates to the nutritional status of animals, is the main hormonal regulator of body growth and development [37,38]. Post-weaning piglets typically experience a decrease in blood IGF-I levels due to significantly lower energy and protein intake [39]. The supplement of pharmacological doses of ZnO in diet can alleviate weaning-related intestinal damage, thereby improving the nutritional status of piglets and the IGF-I level in blood [37]. However, the results of this study showed that 1600 mg Zn/kg ZnO had no effect on serum GH and IGF-I levels in piglets on d 14 after weaning, which is in agreement with previous reports [12,28]. Matteri et al. [39] found that the addition of 1000 mg Zn/kg ZnO in diet did not affect the serum IGF-I level, but the addition of 2500 mg Zn/kg ZnO increased the level of serum IGF-I. Li et al. [40] reported that neither ZnO nor methionine-chelated zinc, at a supplemental dose of 120 mg Zn/kg, could improve the serum GH and ghrelin levels in piglets, which was consistent with our results. These results suggested that 1600 mg Zn/kg of ZnO or 60 mg Zn/kg of Zn-Pro might not be sufficient to ghrelin and GH/IGF-I axis of pigs in the first two weeks after weaning. In addition, the serum IGF-I level of the piglets in the MRP group was significantly greater than that in the other experimental groups on d 14 of experiment, which was consistent with the greater ADG. On d 28 of experiment, the serum GH level of piglets in the ZnO group was significantly greater than that in the Zn-Pro group, indicating that when there was no additional zinc source in the basal diet to meet the nutritional needs of piglets for zinc, 60 mg Zn/kg Zn-Pro alone might not be sufficient to replace ZnO in regard to improving growth performance. There was no difference in ghrelin among all the groups, indicating that the difference in growth performance under this experimental condition was not caused by ghrelin.

The composition and diversity of gut microbiota are important indicators of intestinal microecological stability and inflammation function. The family Erysipelotrichaceae was found to be related to gastrointestinal inflammation [41], so the higher relative abundance of Erysipelotrichaceae (21.56%) in this study may be one of the reasons for the poor growth performance of piglets in the CON group. Some studies have shown that dietary supplementation with pharmacological doses of ZnO can improve gut microbial composition by reducing the proliferation of pathogenic bacteria and increasing the number of beneficial bacteria [29,42,43,44]. In this study, ZnO treatment did not affect the relative abundance of Lactobacilliaceae, but Zn-Pro fed pigs had greater relative abundance of Lactobacilliaceae and *Lactobacillus* in the colonic digesta. *Lactobacillus* is considered to be a beneficial bacterium with various health-promoting effects, such as inhibiting intestinal inflammation, enhancing intestinal barrier function, regulating immune response, and maintaining microbial homeostasis [45]. However, at present, the results of studies about the effects of dietary Zn on *Lactobacillus* are quite different, which may be associated with the application form of Zn and the colonizing site of *Lactobacillus*. Lei et al. [27] reported that ZnO or coated ZnO had no effect on the richness of *Lactobacillus* and *Escherichia coli* in the gut. In addition, Xia et al. [46] reported that dietary supplementation of ZnO or nano-ZnO increased the relative abundance of *Lactobacilli* in the colon, but significantly decreased the abundance of *Lactobacillus* in the ileum. Furthermore, in our study, the dominant bacterium enhanced by MRF was *Prevotella*, whose result was different from Zn-Pro group. The members of the *Prevotella* can degrade polysaccharides of plant cell walls into VFAs that provide energy for gut, improve gut function and reduce diarrhea [16,47]. Research has shown that MOS can improve the intestinal bacterial composition of piglets and increase *Prevotella* in the colon [16]. In addition, *Prevotella_2* has been positively correlated with fecal acetate, propionate, and succinate concentrations [47]. Long et al. [26] also found that the combined use of MOS-rich yeast and ZnO increased the relative abundance of *Prevotellaceae_NK3B31_group* in piglet feces. The *Prevotellaceae_NK3B31_group* has been reported to be inversely correlated with serum IL-6 and TNF-α concentrations and exhibit an anti-inflammatory function [48]. These results suggest that the combined addition of MRF diet can improve gut microbiota composition, especially increasing the abundance of *Prevotella,* which may be part of the reason for its potential as an alternative to ZnO.

Additionally, VFAs produced by the fermentation of hindgut microorganisms also play an important function on reducing intestinal inflammation and improving intestinal health [49]. Previous studies have shown that a high dose of ZnO in diet results in greater concentration of acetic acid in cecal digesta and feces [28,42]. Our result also found that dietary Zn-Pro supplementation at 60 mg Zn/kg markedly increased the acetic acid concentration in the cecum. Acetic acid is mainly transported to the liver through the portal vein and provides energy for peripheral tissues. Additionally, it can promote the diversity of beneficial intestinal microbial by affecting the pH value of intestinal digesta. It has also been shown that propionic acid has a positive effect on gut development and the barrier function of piglets [50]. Long et al. [26] found that feeding MOS-enriched yeast and ZnO increased the concentrations of propionic acid and the total VFA in the feces of piglets. Similarly, MOS has been shown to increase the propionic acid concentration of cecal digesta in broilers [51]. Therefore, the higher propionic acid concentration in the MRP group might be beneficial for improving gut health. 

## 5. Conclusions

In conclusion, the combination of MRF and Zn-Pro reduced the diarrhea rate, and this result might be related to the improved intestinal morphology and microbial structure. In addition, the combined addition of MRF and Zn-Pro may provide benefit to growth performance and nutrient digestibility of weaned pigs. In addition, the fecal zinc excretion was lower with the combined addition of MRF and Zn-Pro, which might decelerate the environmental pollution associated with swine farms. These results suggested that the combination of MRF and Zn-Pro may be an alternative to 1600 mg Zn/kg ZnO.

## Figures and Tables

**Figure 1 animals-12-03407-f001:**
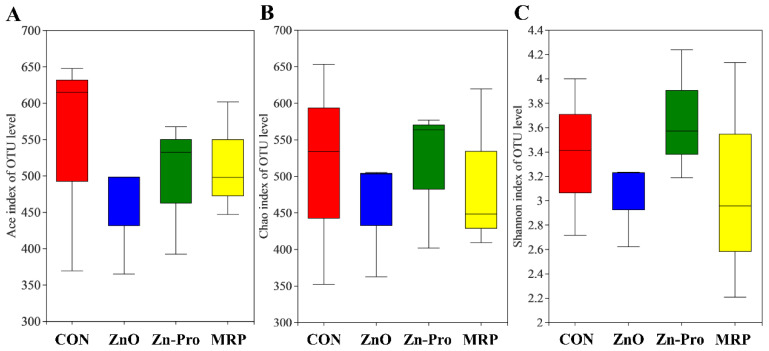
Microbial α-diversity in the feces of weaned pigs on d 28 of experiment. (**A**) Ace index. (**B**) Chao index. (**C**) Shannon index. CON, a basal diet without any Zn source addition; ZnO, basal diet + 1600 mg Zn/kg from zinc oxide; Zn-Pro, basal diet + 60 mg Zn/kg from Zn-Pro; MRP, basal diet + 800 mg/kg MRF + 60 mg Zn/kg from Zn-Pro.

**Table 1 animals-12-03407-t001:** Nutrient levels and composition of the basal diets (%, as-fed basis).

Items	Content
Ingredients	
Corn	60.54
Soybean meal	10.00
Extruded soybean	9.00
Whey power	5.00
Soy protein concentrate	4.00
Fish meal	3.00
Soybean oil	2.50
Glucose	2.00
Dicalcium phosphate	1.14
Limestone	0.87
Salt	0.15
L-Lys HCl, 78%	0.62
DL-Met, 98%	0.11
L-Trp, 98%	0.05
L-Thr, 98%	0.22
Chromic oxide	0.30
Premix ^1^	0.50
Total	100.00
Nutrient levels ^2^	
Metabolizable energy, kcal/kg	3410
Total Ca	0.80
Total P	0.59
Available P	0.40
SID Trp	0.22
SID Met	0.40
SID Thr	0.79
SID Lys	1.35
Analyzed composition	
Dry matter	88.72
Crude protein	18.21
Neutral detergent fiber	86.70
Acid detergent fiber	32.60
Ash	51.40

Note: ^1^ The premix provided the following per kg of diets: vitamin A 12,000 IU, vitamin D_3_ 2500 IU, vitamin E 30 IU, vitamin K_3_ 3.0 mg, vitamin B_12_ 12.0 μg, riboflavin 4.0 mg, pantothenic acid 15.0 mg, niacin 40.0 mg, folacin 0.7 mg, thiamine 1.5 mg, vitamin B_6_ 3.0 m, Mn (as MnO) 30.0 mg, Fe (as FeSO_4_) 70.0 mg, Cu (as CuSO_4_) 20.0 mg, I (as KI) 0.3 mg, Se (as Na_2_SeO_3_) 0.3 mg. ^2^ Calculated values.

**Table 2 animals-12-03407-t002:** Diarrhea rate and growth performance of weaned piglets in the different groups.

Item	Groups				SEM	*p*-Value
	CON	ZnO	Zn-Pro	MRP		
Initial weight/kg (d 28 of birth)	7.71	7.70	7.70	7.69	0.01	0.88
The weight on d 14 (d 42 of birth)	11.62	11.89	11.69	11.90	0.20	0.71
Final weight/kg (d 56 of birth)	17.38^y^	18.19 ^xy^	17.65 ^xy^	18.53 ^x^	0.31	0.07
d 1 to 14 of experiment						
ADG/g	279	299	295	310	11.92	0.37
ADFI/g	440	457	451	459	12.68	0.71
Feed/Gain	1.58	1.53	1.53	1.49	0.03	0.29
Diarrhea rate/%	17.01 ^a^	5.95 ^b^	9.52 ^b^	8.67 ^b^	1.47	<0.01
d 15 to 28 of experiment						
ADG/g	412 ^b^	450 ^ab^	420 ^b^	476 ^a^	12.45	<0.01
ADFI/g	704 ^y^	760 ^x^	714 ^y^	760 ^x^	19.17	0.10
Feed/Gain	1.72	1.68	1.70	1.60	0.04	0.16
Diarrhea rate/%	7.14 ^a^	2.72 ^b^	4.42 ^b^	1.36 ^b^	0.85	<0.01
d 1 to 28 of experiment						
ADG/g	346 ^b^	374 ^ab^	358 ^b^	393 ^a^	9.73	0.02
ADFI/g	572	608	582	609	12.87	0.10
Feed/Gain	1.66	1.62	1.63	1.55	0.03	0.11
Diarrhea rate/%	12.07 ^a^	4.34 ^b^	6.97 ^b^	5.02 ^b^	0.93	<0.01

^a,b^ In the same row, values with different letters mean significant difference (*p* < 0.05); ^x,y^ In the same row, values with different letters mean a tendency difference (0.05 *< p* ≤ 0.10). CON, a basal diet without any Zn source addition; ZnO, basal diet + 1600 mg Zn/kg from zinc oxide; Zn-Pro, basal diet + 60 mg Zn/kg from Zn-Pro; MRP, basal diet + 800 mg/kg MRF + 60 mg Zn/kg from Zn-Pro.

**Table 3 animals-12-03407-t003:** Effects of different treatments on nutrient apparent digestibility and fecal zinc excretion in the different groups.

Item, %	Groups				SEM	*p*-Value
	CON	ZnO	Zn-Pro	MRP		
Apparent nutrient digestibility						
Gross energy	80.66 ^y^	82.01 ^xy^	81.98 ^xy^	82.82 ^x^	0.56	0.09
Dry matter	81.15 ^b^	82.41 ^ab^	82.09 ^ab^	83.07 ^a^	0.45	0.05
Organic matter	83.52 ^b^	84.90 ^ab^	84.72 ^ab^	85.49 ^a^	0.45	0.04
Crude protein	73.81 ^b^	74.95 ^ab^	75.60 ^ab^	76.92 ^a^	0.77	0.05
Neutral detergent fiber	52.61	54.40	55.12	58.37	1.89	0.22
Acid detergent fiber	49.28	47.83	51.90	57.70	2.67	0.10
Feces						
Zn, g/kg DM	0.25 ^c^	12.51 ^a^	0.81 ^b^	0.69 ^b^	0.04	<0.01

^a–c^ In the same row, values with different letters mean significant difference (*p* < 0.05). ^x,y^ In the same row, values with different letters mean a tendency difference (0.05 *< p* ≤ 0.10). Fecal samples were collected on d 25 to 27 of experiment. CON, a basal diet without any Zn source addition; ZnO, basal diet + 1600 mg Zn/kg zinc oxide; Zn-Pro, basal diet + 60 mg Zn/kg from Zn-Pro; MRP, basal diet + 800 mg/kg MRF + 60 mg Zn/kg from Zn-Pro.

**Table 4 animals-12-03407-t004:** Effects of different treatments on serum hormone levels of weaned piglets.

Items	Groups				SEM	*p*-Value
	CON	ZnO	Zn-Pro	MRP		
d 14 of experiment						
Growth hormone/(ng/mL)	3.90	4.41	3.80	4.40	0.23	0.16
Insulin-like growth factor-1/(ng/mL)	139.30 ^b^	136.36 ^b^	128.27 ^b^	179.05 ^a^	8.22	<0.01
Ghrelin/(ng/mL)	61.79	70.92	57.61	70.58	4.38	0.11
d 28 of experiment						
Growth hormone/(ng/mL)	6.76 ^ab^	7.18 ^a^	5.38 ^b^	5.99 ^ab^	0.44	0.04
Insulin-like growth factor-1/(ng/mL)	246.70	267.30	260.61	278.24	9.85	0.18
Ghrelin/(ng/mL)	81.96	87.48	78.55	105.34	8.56	0.16

^a,b^ In the same row, values with different letters mean significant difference (*p* < 0.05). CON, a basal diet without any Zn source addition; ZnO, basal diet + 1600 mg Zn/kg zinc oxide; Zn-Pro, basal diet + 60 mg Zn/kg from Zn-Pro; MRP, basal diet + 800 mg/kg MRF + 60 mg Zn/kg from Zn-Pro.

**Table 5 animals-12-03407-t005:** Effects of different treatments on intestinal morphology in the different groups on d 28 of experiment.

Items	Groups				SEM	*p*-Value
	CON	ZnO	Zn-Pro	MRP		
Jejunum						
Villus height/(μm)	292	289	284	313	22.4	0.82
Crypt depth/(μm)	180	159	159	157	9.00	0.26
Villus height/Crypt depth	1.63	1.84	1.84	2.02	0.21	0.65
Ileum						
Villus height/(μm)	220 ^y^	240 ^xy^	227 ^xy^	255 ^x^	9.36	0.09
Crypt depth/(μm)	180	159	154	158	8.70	0.21
Villus height/Crypt depth	1.24	1.52	1.49	1.63	0.12	0.18

^x,y^ In the same row, values with different letters mean a tendency difference (0.05 *< p* ≤ 0.10). CON, a basal diet without any Zn source addition; ZnO, basal diet + 1600 mg Zn/kg zinc oxide; Zn-Pro, basal diet + 60 mg Zn/kg from Zn-Pro; MRP, basal diet + 800 mg/kg MRF + 60 mg Zn/kg from Zn-Pro.

**Table 6 animals-12-03407-t006:** Effects of different treatments on volatile fatty acid concentrations in the different groups on d 28 of experiment.

Items, mg/g	Groups				SEM	*p*-Value
	CON	ZnO	Zn-Pro	MRP		
Cecum						
Lactic acid	0.64	0.44	1.37	0.58	0.51	0.58
Acetic acid	4.59 ^c^	5.28 ^ab^	5.69 ^a^	4.94 ^bc^	0.18	<0.01
Propionic acid	2.83 ^b^	3.37 ^ab^	2.99 ^b^	3.73 ^a^	0.18	0.01
Butyric acid	1.25	1.26	1.57	1.31	0.16	0.50
Valeric acid	0.07	0.08	0.11	0.04	0.03	0.47
Total volatile fatty acids	9.59 ^b^	10.47 ^ab^	11.77 ^a^	10.62 ^ab^	0.42	0.02
Colon						
Lactic acid	0.31	0.21	0.07	0.02	0.08	0.13
Acetic acid	5.18	5.43	5.50	5.15	0.17	0.37
Propionic acid	2.92	3.44	3.11	3.45	0.27	0.56
Butyric acid	1.40	1.70	1.76	1.84	0.17	0.31
Valeric acid	0.20	0.21	0.21	0.13	0.04	0.54
Total volatile fatty acids	10.12	11.19	10.82	10.42	0.53	0.54

^a–c^ In the same row, values with different letters mean significant difference (*p* < 0.05). CON, a basal diet without any Zn source addition; ZnO, basal diet + 1600 mg Zn/kg zinc oxide; Zn-Pro, basal diet + 60 mg Zn/kg from Zn-Pro; MRP, basal diet + 800 mg/kg MRF + 60 mg Zn/kg from Zn-Pro.

**Table 7 animals-12-03407-t007:** Effects of different treatments on the relative abundance of main microbiota in colonic digesta of weaned piglets on d 28 of experiment.

Items	Groups				SEM	*p*-Value
	CON	ZnO	Zn-Pro	MRP		
Phylum level						
Firmicutes	78.76	84.52	89.49	67.16	5.53	0.11
Bacteroidetes	19.23	13.59	6.39	30.10	5.39	0.11
Family level						
Ruminococcaceae	26.72 ^x^	8.66 ^y^	17.50 ^xy^	16.29 ^xy^	4.81	0.09
Clostridiaceae	11.36	30.63	1.70	19.20	9.21	0.15
Prevotellaceae	15.28 ^ab^	12.04 ^ab^	0.85 ^b^	27.45 ^a^	5.09	0.05
Lactobacillaceae	1.46 ^y^	19.38 ^xy^	27.73 ^x^	3.30 ^y^	8.80	0.08
Erysipelotrichaceae	21.56	1.54	2.47	6.47	5.40	0.13
Lachnospiraceae	3.50 ^y^	8.27 ^xy^	19.78 ^x^	2.77 ^y^	3.75	0.09
unclassified_o_Lactobacillales	0.01 ^b^	0.81 ^ab^	2.06 ^a^	0.01 ^b^	0.60	0.04
Genus level						
*Clostridium_sensu_stricto_1*	11.21	28.51	1.52	19.00	8.77	0.15
*Lactobacillus*	1.46	19.38	27.73	3.40	8.80	0.13
*Prevotella_2*	4.76 ^xy^	3.49 ^xy^	0.03 ^y^	13.39 ^x^	6.66	0.09
*Ruminococcaceae_UCG-005*	8.30 ^x^	0.73 ^y^	0.83 ^y^	1.98 ^xy^	1.08	0.06
*Prevotella_9*	2.95	2.42	0.27	4.06	1.33	0.17
*Prevotellaceae_NK3B31_group*	2.04 ^xy^	2.29 ^xy^	0.15 ^y^	3.03 ^x^	0.84	0.08

^a,b^ In the same row, values with different letters mean significant difference (*p* < 0.05). ^x,y^ In the same row, values with different letters mean a tendency difference (0.05 *< p* ≤ 0.10). CON, a basal diet without any Zn source addition; ZnO, basal diet + 1600 mg Zn/kg zinc oxide; Zn-Pro, basal diet + 60 mg Zn/kg from Zn-Pro; MRP, basal diet + 800 mg/kg MRF + 60 mg Zn/kg from Zn-Pro.

## Data Availability

Data sharing is not applicable to this article.

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
