# Peer review of "Effects of Protein-Chelated Zinc Combined with Mannan-Rich Fraction to Replace High-Dose Zinc Oxide on Growth Performance, Nutrient Digestibility, and Intestinal Health in Weaned Piglets"

_animals, 2022, doi:10.3390/ani12233407_

Round 1

Reviewer 1 Report

Early weaning is an important period that affect the growing and finishing phases in pigs. Using zinc supplementation to improve intestinal health is of utmost importance. Although, this study has merit, I think the author need to work on making the paper more readable. The result section is poorly written. In addition, the discussion needs to the refined because some paragraphs are just the review of the literature and don’t really mention and discuss the results of the current study. Following are my comments and suggestions to improve the content and the presentation of this paper.

Introduction

Line 41: “show a trend toward being greater” should be more precise. Please, replace by tended to be greater than.

Line 54: replace mother by sow.

Lines 67-70: need references

Line 81. “A mannan-rich fraction…as the second generation” The end of the sentence doesn’t make sense and should be revised.

Line 91: what about intestinal microbiota?

 Materials and methods

Line 127: which equipment was used for the grounding the feces? In addition, replace reserved doesn’t have the same meaning as stored. Please, replace by stored.  

Line 130: revise the syntax for this sentence to make it more readable.

Line 131: I don’t think that “chyme” is the appropriate word here. In the colon and the ceca, we don’t we have chyme anymore. Chyme is the content that flows from the stomach into the duodenum. Should simply use colon and cecal content.

Line 150: should read fix in formalin not at formalin.

Line 151: should be, was stained using HE, not by.

Line 52: should be measured, using image analyzer, not by.

Line 142: remove an extra of after V3-V4 region.

Line 167: no need to repeat ghrelin in parenthesis since it is only one work.

Line 172: should be pen as experimental unit.

Line 174-177: I suggest the authors split this into 2 or 3 sentences to make it simple and more readable.

Line 176: for example, LSMEANS method multiple comparison. This need to be revised.

Line 176: spell out NSK

Line 179: I don’t think “trendy” means a tendency. I suggest the author use a well-defined term. Use  “tendency” instead of “trendy significance”.

This should apply for the rest of the paper: use tended to (see line 223-236 as example).

Results

I the result section is so poorly written. The authors need to revise this section to capture what they have put in the tables.

I suggest citing the table each time or put at the beginning of each result paragraph in a separate sentence where to find the data. For example: the performance and diarrhea data are presented in table 2 or just cite the table each time you describe a result (see line 223-224 as example).

Line 184: “in addition, during the 2 phase (d 15 to 28) and whole period”. This is confusing. First what is whole period? I suggest using the same denominations in the tables.

Line 185: this is not what I read in the table.

Line 188: If you have defined tendency in the statistical analysis why use numerically increased? Just say that X tended to be greater than Y.

Tables

If you have defined tendency, you need to add the mean separation using different letters. For example, x,y,z etc.. otherwise, I cannot read and interpret your tables.  For example, Final weight and ADFI in table 2. Please revise all the tables accordingly.

Revise tables formatting presentation to remove extra lines and adjust the rows and columns. 

Line 208; “however, gut morphology measures showed no significant different …… all the four groups. First, “different” should be differences. Second, this sentence contradicts the previous one because villus height is also gut morphology. The authors should be precise by mentioning which parameters they are referring to.

All the tables should stand alone like the figure 1. The authors should mention the treatments in the table footnotes.

Discussion

Line 263: the combination of what?

Aline 293: replace closely by close

Paragraph line 293-311 doesn’t flow well. The authors have talked about nutrient digestibility, intestinal morphology, intestinal microbiota, and fecal zinc. I suggest rearranging and discussing these points separately or at least show some transition to help understand the message.

Line 304: what improvement of the intestinal epithelium? What prebiotic?  

The use of whish throughout the discussion is confusing. Which following a comma generally refers to the word directly before the comma. Please, revise all the “whish”.

Line 341-342 should be moved to the end of the paragraph to serve as a conclusion.

This paragraph 346-374 is more like a literature review not a discussion. I don’t see any discussion of the microbiota data. The only time a result of this study is mentioned is on line 363. Please revise to place your results in the current literature, comparing it with previous reports, and stating their significances.

Line 374: in conclusion, while..., but beneficial. For this sentence to make sense you should remove the “but”.

Reviewer 2 Report

The paper deals with the ever-present issue of feed additives for piglets, showing health-promoting effects and reducing the occurrence of diarrhea in the peri-weaning period. An additional aspect is the undoubted necessity to search for an alternative to zinc oxide used in medicated feeds.

Please clarify on the following points:

1 The premix contains Zn (as ZnO) 75 mg. How to understand the statement basal diet + experimental factor (lines: 101-103). So, how many Zn (total value) were in the each experimental groups ?

The experimental factor was to different the dose and source of zinc. Therefore, it is not clear what the total level of Zn was in the doses. I propose to show the level of Zn measured in the mixtures in all experimental groups.

2. On what day of life were the piglets weaned and on what day of animal's life was the experiment completed ? Table 2 says: initial and final body weight. Please complete which is the day of life of the animals.

3. Were the piglets weighed on day 14 ? The body weight of piglets on day 14 should be shown in Table 2.

4. Why is it written: The feeding experiment lasted for 28 days, divided into two period of 14 days each. How did these periods differ ? Why was this day (the 14th day"s of life) distinguished ? After all, the animals received the same feed for 28 days.

5. Under the table 1 it is written: CP was a measured value, while the others were calculated values. On the other hand, in the paper we can read that fecal samples and feed were analyzed .... (line 157). This statement is not clear to me. Please clarify. In my opinion, it is better to provide measured values in the table than theoretically calculated ones.

6. The header of the tables shows the variation in zinc source (CON, ZnO, Zn-Pro and MRP). I wonder if you should add a row and also show the variation in the levels of Zn used.

7. It is difficult to analyze Table 3 without showing the Zn values in the feed (I wrote about this earlier - 1st point).

8. Have you measured Zn in the blood ?

9. It is a pity that the blood was not taken before the start of the experiment and the selected parameters were not measured in it. Seeing the values at the start of the experiment would have made it easier to infer changes after the administration of the experimental factors.

10. I propose to rename tables 3 to 7. It is written: Effects of Z-Pro and MRF on ... Why only Zn-Pro nad MRF : How about: effect of different source and level of Zn on ....

English was not reviewed.

 All the best and stay safe

Reviewer 3 Report

The objective of this paper was to evaluate substitutes for zinc ocide in nursery pigs. The authors revaluated zinc protein chelate or in combination with mannan rich fraction (MRF) on nursery pig performance, intestinal morphology, AATD, serum IGF-1 an, ghrelin and growth hormone, and fecal microbiota. Four treatments containing 7 pens (6 pigs/pen, split equally by sex) were utilized in a 28 day study. The treatments were: 1) Control, zinc oxide (1600 ppm Zn), Zn-Pro and Zn-Pro-MRF. The authors conclude that Zn-Pro alone did not improve growth performance, but changed the microbiota. MRF treatment benefited growth, morphology and IGF-1. Further this treatment had lower diarrhea scores. The authors present a nice paper that is relevant to the swine nutrition with the ban on high zinc diets and this work warrants publication. I would encourage the authors to address the following comments.

Comments:

1.      Line 18: specify Zn in the form of zinc oxide

2.      Line 19: antimicrobial resistance.

3.      Line 20: harm health of animal and humans? Misleading. Please clarify or re-state differently.

4.      Line 52: Weaning is not a technology. Please re-phrase.

5.      Throughout the manuscript, please be clear on zinc or zinc oxide. This issue in the industry is the concentration of zinc (pharmacological/therapeutic) in the form of zinc oxide.

6.      Line 63: Please also state these negative effects to the pig.

7.      Line 66: Delete “Besides”.

8.      Line 66: What is excessive? Please provide concentration range.

9.      Line 68: Zn not ZnO.

10.   Line 70 and 71: Zn not ZnO.

11.   Line 71: How does zinc prevent growth retardation? This is misleading. It aids weaning transition to help with enhanced growth.

12.   Line 77: what are harmful bacteria? Please specify.

13.   Line 80: What is a “massive discharge of zinc”? I assume you are referring to poor retention rates. If so, cite a nursery study that has examined ATTD of Zn.

14.   Line 87: Zn emissions? Excretion.

15.   Line 96: Zn.

16.   Line 98: Genetics of the pigs? Age? Freshly weaned? Vaccines? Health status? All must be reported.

17.   Line 108: indicator for what?

18.   Line 121: Diarrhea incidence – are you referring to the % of pigs with a score of 4 and 5 or just 5? Clarify.

19.   Line 129: euthanized and necropsied, and then ….

20.   Line 130: What is the location of sampling within the jejunum and ileum? Flushed with what? Specify.

21.   Line 131: chyme is stomach contents expelled into the duodenum. Please clarify.

22.   Line 140-166: More details of methods needed or clearer citations of method conditions.

23.   Line 167: Serum growth hormone ….

24.   Line 169: What type of assay kit (EIA, RIA)? How and what was read and in what instrument? Details.

25.    Line 172: Pen served as the experimental unit for ….

26.   Line 177: More details needed on the microbiota analysis. How was this down. How many reads? How was the data quality assessed?

27.   Line 179, 186 and elsewhere: delete “trendy” throughout and change to “tended”

28.   Tables all need better formatting.

29.   Table 5: villus height and crypt depths to no decimal places. i.e. 292 um. Keep ratios as is.

30.    Table 6: units missing?

31.   Line 222: Delete “The indicators such as”

32.   Line 247: Expand on this. Cite data reporting feed intakes. Stresses such as what?

33.   Line 248: Ghrelin and feed intake references needed. There is recent work highlight Zn and stomach ghrelin and feed intake in pigs. I would encourage the authors to review and cite.

34.   Line 254: How is MOS potentially alleviating weaning stress?

35.   Line 262: But same between ZnO and MRF.

36.   Line 258: emissions?

37.   Line 274: Not clear on this sentence here. Re-phrase. \

38.   Line 275: How does blood Zn reduce growth retardation?

39.   Line 277: Logic should also apply to the MRF group! Same Zn load. Please address.

40.   Line 288: references need. Which pathogens specifically?

41.   Discussion: What is causing the diarrhea in the current study? This needs to be discussed.

42.   Line 290: But riches/diversity not different herein (Figure 1). Please explain.

43.   Line 293: How related?

44.   Line 294: What type of damage? How?

45.   Line 309: Discuss in terms of Zn retention.

46.   Discussion: Why was Ghrelin measured and not discussed? Please address.

47.   Line 349: These references do not show specific pathogen decreases. Find papers that do.

48.   Line 355: Greater relative abundance of Lactobacillus is good, then is it an issue the MRF pig shad the lowest abundance (Table 7)? Your 16S data would suggest that composition is not the issue. Please address in discussion.

49.   Discussion: The authors are encouraged to discuss how the treatments are an alternative to ZnO and not just to the Control.

50.   The introduction and Discussion: What are the authors specifically trying to fine an alternative for with regard to ZnO. Is it feed intake, growth, diarrhea incidence, anti-pathogenic etc… This is unclear.

51.   How is the combo MRF diet providing equal performance? This needs stronger discussion, in particular mode of action.

Round 2

Reviewer 1 Report

The authors have made significant efforts to respond to my comments and suggestions. Few minor grammatical and syntax issues need to be addressed. Please, see my comments below.

Line 60: should be in order to mitigate…..improve

Line 86: should be has been launched instead of has launched

Line 89: review the beginning of this sentence ”combined addition”

Line 130: should be ” and then passed” not “to pass”

Line 135: what is it? Plasma or serum!

Line 137: “and then removed the intestine” should be “and then the intestine was removed”

Line 198: use tendency instead of trendy significance

Line 208: should be tended not trended.

Line 220: separate his into 2 sentences for more clarity because the second part of the sentence (and trended to increase ….) is not clear. Again, we say trend, but we use tendency (not trendy) or tended (not trended).

Line 239: what about the ileum results? The p value is 0.09.  

Line 397: replace flora by bacteria

Line 406: enhanced by of MRF. (Enhanced by what?)

Line 421-423: the sentence need revision. “Which these results were similar to” doesn’t make sense reading

Line 424: additionally, already means also (remove one)

Line 434: which might be contributed. (Revise this to read better)

Line 439-440: this is an assertion from your study and verb tense is not appropriate. The results suggest that …..may be an ….
